# Pancreatic cancer prognosis is predicted by an ATAC-array technology for assessing chromatin accessibility

S. Dhara[1,9], S. Chhangawala [2,3,9], H. Chintalapudi[1], G. Askan[4], V. Aveson[4,5], A. L. Massa [1], L. Zhang[4], D. Torres[1], A. P. Makohon-Moore [4], N. Lecomte[4], J. P. Melchor[4], J. Bermeo[4], A. Cardenas III[4], S. Sinha[4], D. Glassman [4], R. Nicolle [6], R. Moffitt [7], K. H. Yu[4], S. Leppanen[8], S. Laderman[8], B. Curry[8], J. Gui[1], V. P. Balachandran [4], C. Iacobuzio-Donahue [4], R. Chandwani[5], C. S. Leslie [3✉] & S. D. Leach[1✉]

Unlike other malignancies, therapeutic options in pancreatic ductal adenocarcinoma (PDAC) are largely limited to cytotoxic chemotherapy without the benefit of molecular markers predicting response. Here we report tumor-cell-intrinsic chromatin accessibility patterns of treatment-naïve surgically resected PDAC tumors that were subsequently treated with (Gem)/Abraxane adjuvant chemotherapy. By ATAC-seq analyses of EpCAM[+] PDAC malignant epithelial cells sorted from 54 freshly resected human tumors, we show here the discovery of a signature of 1092 chromatin loci displaying differential accessibility between patients with disease free survival (DFS) < 1 year and patients with DFS > 1 year. Analyzing transcription factor (TF) binding motifs within these loci, we identify two TFs (ZKSCAN1 and HNF1b) displaying differential nuclear localization between patients with short vs. long DFS. We further develop a chromatin accessibility microarray methodology termed "ATAC-array", an easy-to-use platform obviating the time and cost of next generation sequencing. Applying this methodology to the original ATAC-seq libraries as well as independent libraries generated from patient-derived organoids, we validate ATAC-array technology in both the original ATAC-seq cohort as well as in an independent validation cohort. We conclude that PDAC prognosis can be predicted by ATAC-array, which represents a low-cost, clinically feasible technology for assessing chromatin accessibility profiles.

[1] Dartmouth Geisel School of Medicine and Norris Cotton Cancer Center, Hanover, NH, USA. [2] Weill Cornell Graduate School of Medical Sciences, New York, NY, USA. [3] Computational Biology Program, Memorial Sloan Kettering Cancer Center, New York, NY, USA. [4] David M. Rubenstein Center for Pancreatic Cancer Research, Memorial Sloan Kettering Cancer Center, New York, NY, USA. [5] Weill Cornell Medicine, New York, NY, USA. [6] Programme Cartes d'Identité des Tumeurs, Ligue Nationale Contre Le Cancer, Paris, France. [7] Stony Brook University, Stony Brook, NY, USA. [8] Agilent Technologies Inc., Santa Clara, CA, USA. [9] These authors contributed equally: S. Dhara, S. Chhangawala. ✉email: cleslie@cbio.mskcc.org; sdl@dartmouth.edu

Prognostic precision is pivotal to therapeutic decision making. Pancreatic ductal adenocarcinoma (PDAC) patients with the limited local disease and with no detectable metastasis typically have their primary tumor surgically resected and subsequently receive adjuvant chemotherapy. Approximately 50% of these resected PDAC patients have a poor prognosis, with disease recurring within 1 year of surgery in spite of apparently complete tumor removal (R0 margin-negative resection) and adjuvant chemotherapy. An additional 30-35% of patients recur within 2-5 years, but a small subset (14%), is characterized by prolonged disease-free survival (DFS) of 5–10 years[1–3]. Recurrence in spite of adjuvant chemotherapy may be due to the presence of undetected chemotherapy-refractory micro-metastatic lesions, while non-recurrence may mean that the tumor never metastasized or that micro-metastatic lesions responded well to the adjuvant chemotherapy. Previous literature including large-scale efforts from both the TCGA and the ICGC have reported genomic and transcriptomic analyses on resected PDAC bulk tumors[4–7], with follow-up studies ongoing to derive clinically actionable leads[8]. Complementing these large-scale studies of differential gene expression in PDAC, we sought to map in parallel the epigenetic heterogeneity of this disease. Considering the variable epithelial cellularity of PDAC tumor microenvironment[9], rather than relying on bulk tumor material, we decided to investigate the tumor-cell-intrinsic epigenetic profile of resected PDAC. By analyzing genome-wide chromatin accessibility patterns by Assay for Transposase-Accessible Chromatin sequencing (ATAC-seq)[10] of EpCAM-sorted PDAC malignant epithelial cells from 54 surgically resected PDAC tumors we identified 121,697 open chromatin peaks and discovered 1092 regions that are differentially represented between patients with DFS < 1 year and patients with DFS > 1 year. We further identified and validated two transcription factors (TFs) HNF1b and ZKSCAN1, with differential nuclear localization between poor and good prognosis tumors. We also developed "ATAC-array" technology—the only microarray that reads chromatin accessibility patterns—as a potential diagnostic tool that may be utilized in PDAC clinic. Combining ATAC-array and HNF1b together we segregate resected PDAC patients into good, intermediate, and poor prognosis groups with greater than 7-fold differences in DFS.

## Results

**PDAC malignant epithelial cellularity is variable.** We hypothesized that epigenetic differences at the level of chromatin accessibility, potentially linked to distinct differentiation states, might distinguish rapidly recurrent from non-recurrent tumors. To test our hypothesis, we sought to characterize tumor-cell intrinsic chromatin accessibility patterns from a prospective cohort of treatment-naïve, surgically resected tumors from 54 PDAC patients at Memorial Sloan Kettering Cancer Center. Consistent with the known issue of variable neoplastic cellularity in PDAC, an initial histopathological survey of frozen archival tissues in our repository ($n = 120$) found variable epithelial contents ranging from 0 to 90%, with median cellularity of 40% (Supplementary Fig. 1a). To overcome this variability, we optimized the sorting of PDAC malignant cells from freshly resected tumors using EpCAM antibody-conjugated magnetic beads (Supplementary Fig. 1b). We collected both EpCAM$^+$ and EpCAM$^-$ cells from each of the tumors and confirmed effective enrichment of malignant epithelial cells by comparing 15 variant allele frequencies (VAFs) of canonical pancreatic cancer driver genes, including six variant alleles for *KRAS* and nine variant alleles for *TP53* (Supplementary Fig. 1c), where each of the variant alleles is denoted by different colored lines comparing EpCAM$^+$ and EpCAM$^-$ subpopulations of the same tumor

collected from each patient. We observed that the VAFs of *KRAS* and *TP53* were both dramatically higher in the EpCAM$^+$ cells than in the EpCAM$^-$ cells ($P < 0.001$, t-test), confirming effective enrichment of malignant epithelial cells in the EpCAM$^+$ subpopulation. This enrichment was further confirmed by transcriptome analysis using Quant-Seq[11], which demonstrated high-level expression of epithelial genes in the EpCAM$^+$ subpopulation, with the corresponding expression of immune cell and collagen genes in the EpCAM$^-$ subpopulation (Supplementary Fig. 2 and Supplementary Data 1).

**ATAC-seq analysis on sorted PDAC malignant cells.** We then performed Assay for Transposase-Accessible Chromatin sequencing (ATAC-seq) analysis[10] on the EpCAM$^+$ cells to interrogate genome-wide chromatin accessibility and associated differentially accessible TF-binding sites. After initial quality control (described in Methods and in Supplementary Fig. 3), we assembled a global atlas of 121,697 chromatin peaks, where each peak was reproducible in replicate ATAC-seq libraries for at least two patients. We performed saturation analysis to estimate incremental new peak discovery associated with stepwise increases in sample size and confirmed that a sample size of $n = 40$ approached saturating coverage (Supplementary Fig. 4a).

**Discovery of 1092-peak chromatin accessibility signature.** Follow-up clinical data were available for 36 out of 40 patients included in the atlas (see remarks in Supplementary Data 2 and Supplementary Fig. 4b). 19 out of 36 patients were at least 365 days post-treatment, among whom 9 patients (47.4%) had recurred (DFS < 1 year, referred to as the recurrent group), and 10 patients had no recurrence (DFS > 1 year; maximum of 660 days, referred to as the non-recurrent group). The latter group, however, was expected to represent a mixture of long-term survivors and others who would likely recur in 2–5 years. For the discovery analyses, we excluded 3 patients who did not receive any adjuvant chemotherapy, leaving 16 patients (6 recurrent and 10 non-recurrent) in the initial training set. We then used a multi-factor generalized linear model to identify significant differential chromatin accessibility events between the recurrent versus non-recurrent groups, while controlling for the effects of nuclear read depth and margin status. In this regard, nuclear read depth displayed an inverse correlation with the mitochondrial read depth in all the ATAC-seq libraries, eliminating the need to consider both in the analysis (Spearman $\rho = -0.273$, 95% CI $-0.439$ to $-0.088$, $P = 0.0044$, $n = 108$). We found 1092 peaks to be differentially accessible (absolute log$_2$ fold change > 1 and FDR-adjusted $P < 0.001$) between recurrent and non-recurrent patients (Fig. 1a, Supplementary Data 3). We applied Cox regression to evaluate the confounding effect of age, sex, and cellularity on DFS. We plotted the Kaplan–Meier curve (Supplementary Fig. 5a log rank $P < 0.0001$, HR 0.1579, 95% CI of HR 0.02877 to 0.8665, median DFS recurrent 236.5 and non-recurrent 927.5 days) with a median 4.15 (min = 3.18, max= 4.75) years of follow-up on the discovery set patients ($n = 16$) and adjusting for age, sex, and cellularity (*KRAS* variant allele frequency)[12,13]. Neither the variant allele frequencies for *KRAS* and *TP53* nor levels of *EpCAM* and *KRT19* gene expression were significantly different between the 6 recurrent and 10 non-recurrent patients (Supplementary Fig. 5b, c), confirming no confounding effect of epithelial cellularity on the discovery of our differentially accessible 1092 chromatin peak signature.

Interestingly, the expression of genes associated with differentially closed peaks was significantly downregulated in the EpCAM$^+$ cells of the recurrent versus non-recurrent tumors ($P < 2.5 \times 10^{-9}$, Kolmogorov–Smirnov test), but expression of genes near

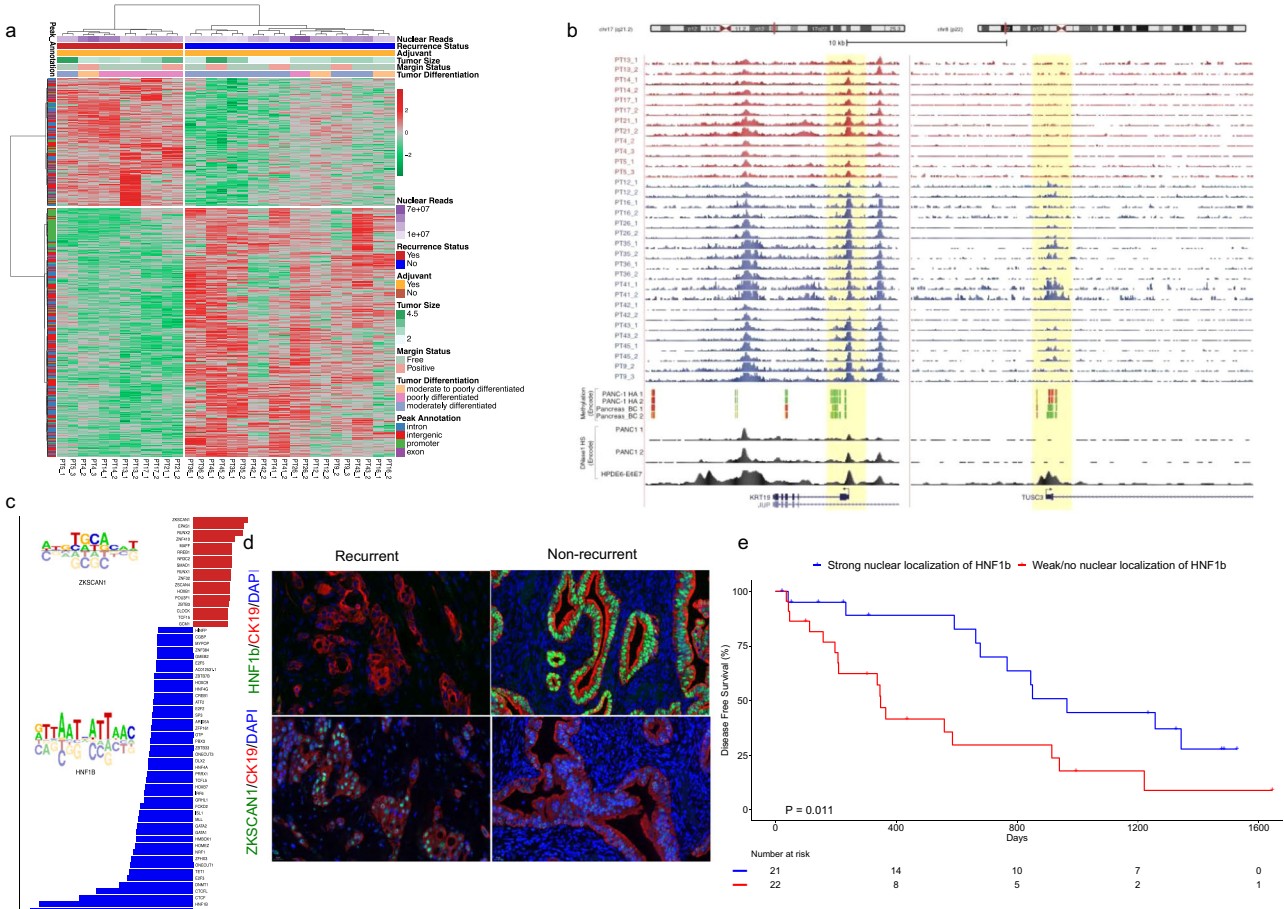

**Fig. 1 Differential chromatin accessibility signature and transcription factor nuclear localization predict prognosis of resected PDAC. a** Differential ATAC-seq peaks in recurrent versus non-recurrent patients (Source data are provided as a Source Data file). **b** Upper panel depicts genome browser tracks showing ATAC-seq peaks at *KRT19* and *TUSC3* gene loci for duplicate samples from individual patients (PT); lower panel shows corresponding ENCODE methylation signals from Panc-1 and a normal pancreatic (BC) cell lines, (green denotes hypomethylation and red denotes hypermethylation) and DHS peak from Panc-1 and HPDE6-E6E7 cell lines. Highlighted yellow regions denote the promoter peaks of both the gene loci. All the track lines have the same *Y* axis limits and that the peak height is normalized by the total read depth of each sample. **c** Regression coefficients showing enrichment of transcription factor-binding sites in recurrent (red) and non-recurrent (blue) patients. **d** Representative images of TMA staining of HNF1b (upper panel) and ZKSCAN1 (lower panel) in recurrent and non-recurrent patients (TMA n = 43 with three replicated cores in each patient), Scale bars are 20 µM as displayed at the left bottom corners of all the micrographs. **e** Kaplan–Meier curve of the patients with and without nuclear localization of HNF1b (log-rank (Mantel–Cox) test P = 0.011, HR 2.64, 95% CI 1.213 to 5.775). (Source data are provided as a Source Data file).

differentially open peaks was not significantly upregulated compared to the background of genes near unchanged peaks (Supplementary Fig. 6a). Figure 1b shows the putative promoter region of *TUSC3* gene, which was less accessible in the recurrent tumors, consistent with its mRNA expression (Supplementary Fig. 6b). The promoter region of *KRT19* (as internal control), a marker gene for pancreatic ductal differentiation, showed no difference in accessibility and no differences in mRNA expression between groups. We interrogated these loci in the ENCODE database for a pancreatic cancer cell line (Panc-1) and two normal pancreatic cell lines (HPDE, pancreas BC). The *TUSC3* promoter region displayed hypermethylation in Panc-1 and hypomethylation in pancreas BC (Fig. 1b), whereas hypomethylation at the *KRT19* region was visible in both cell lines. Also, there was no DNase 1 hypersensitive site (DHS) detected at the *TUSC3* promoter in Panc-1, while it was clearly detected in HPDE.

**Transcription factor-binding motif search and validation.** Next, we performed an in silico search for transcription factor (TF) binding motifs across the whole atlas and used ridge

regression to find TF-binding sites that predicted differential accessibility. We identified 61 TFs whose motifs were differentially open in recurrent (17 motifs) and non-recurrent (44 motifs) patients (Fig. 1c). To test if the discovered transcription factors (TFs) were actually localized to cell nuclei of the respective tumors, we selected two TFs among the top hits, ZKSCAN1 (motifs differentially open in recurrent patients) and HNF1b (motifs differentially open in non-recurrent patients), and performed immunohistochemistry (IHC) and immunofluorescence (IF) staining on tissue microarrays (TMAs) prepared from triplicate cores of formalin-fixed paraffin blocks of 43 out of 54 tumors, followed by a blinded subjective scoring (0–3 scale) of the IHC results and quantitative assessment of the IF results. Zinc finger with KRAB and SCAN domains 1 (*ZKSCAN1*, also known as *ZNF139*) is a zinc-finger transcription factor, and its overexpression in gastroesophageal adenocarcinoma at the esophagogastric junction has been reported[14,15]. Hepatocyte nuclear factor 1b (HNF1b) is a known prognostic biomarker in many cancer types including PDAC, where it plays a role suppressing tumor progression[16–18]. We have also searched for motifs of HNF1b and ZKSCAN1 to identify putative binding sites

throughout our global atlas of open chromatin peaks (121,697 peaks) and found 9613 and 5339 hits for HNF1B and ZKSCAN1 respectively. The list of nearest genes for these TF-binding motif hits is displayed in Supplementary Data 4, and Supplementary Data 5. We considered nuclear staining to be a positive indicator of nuclear localization of these TFs (Supplementary Fig. 6c, d). The nuclear staining patterns of HNF1b and ZKSCAN1 in representative recurrent (Fig. 1d (upper right) and (lower right), respectively) and non-recurrent patients (Fig. 1d ii and iv, respectively) are shown. Kaplan–Meier analysis (Fig. 1e) showed significant segregation of DFS between patients with strong nuclear localization of HNF1b versus patients with weak/no nuclear localization of HNF1b (log-rank (Mantel–Cox) test $P = 0.011$, Gehan–Breslow–Wilcoxon test, $P = 0.006$, HR 2.64, 95% CI 1.213 to 5.775, median DFS 965 and 348 days respectively).

**Validation of HNF1b and ZKSCAN1 on independent PDAC cohort.** We further tested nuclear localization of HNF1b and ZKSCAN1 by multiplex immunofluorescence staining using a TMA representing tumors from an independent archival PDAC cohort ($n = 97$), where the short-term ($n = 45$ with median overall survival (OS) 6 months) and the long-term survivors ($n = 52$ with median OS 6 years) had already been preselected[2]. We observed only rare cells with HNF1b nuclear staining in the tumors of short-term survivors, but many in long-term survivors (Fig. 2a (upper left) and (upper right) respectively). By quantitative estimation of the proportion of nuclear-positive cells, the long-term survivors showed a 52-fold increase in HNF1b nuclear localization compared to the short-term survivors (Fig. 2b (upper left). Conversely, ZKSCAN1 displayed a 5.3-fold lower rate of nuclear localization in long-term survivors compared to short-term survivors (Fig. 2b (lower left). For both TFs, a simple determination of total area staining positive was much less

discriminative (Fig. 2b (upper and lower right) suggesting that the nuclear localization of these TFs, rather than their overall expression, is more predictive of recurrence. These studies demonstrate that the expression and localization of HNF1b protein, a transcription factor identified through unbiased assessment of chromatin accessibility, is different between samples with short and long DFS.

**Development of ATAC-array technology.** As a means to simplify the assessment of chromatin accessibility signatures to the point of clinical utility, we developed a microarray approach that we termed "*ATAC-array*", where the accessible regions from the differential chromatin accessibility signatures were arrayed on glass slides and then hybridized with fluorescent-labeled ATAC libraries. In developing this technology, we reasoned that since the ATAC libraries contain only the accessible regions (unlike the whole-genome libraries), hybridization of these ATAC libraries with the ATAC-array probes would provide specific signal intensities corresponding to the relative abundance of informative accessible regions as represented in each library (explained in the method section and in the schematic diagram of Fig. 3). We compared ATAC-array results (normalized hybridization intensities) side-by-side with ATAC-seq (normalized read-counts) for 932 of the 1092 regions that were covered by 5227 Agilent probes, and found a significant correlation between these two technologies in each patient ($n = 36$, Spearman $\rho$ min=0.5, median=0.65, and max=0.77), as shown in Fig. 4a for a representative patient PT17 (Spearman $\rho = 0.6615$, 95% CI 0.6226 to 0.6971, $P < 0.0001$, number of pairs 931).

**Derivation of ATAC-array prognosis score.** For each ATAC-array analysis, we measured four hybridization intensity distributions relative to distinct probe sets and summarized these

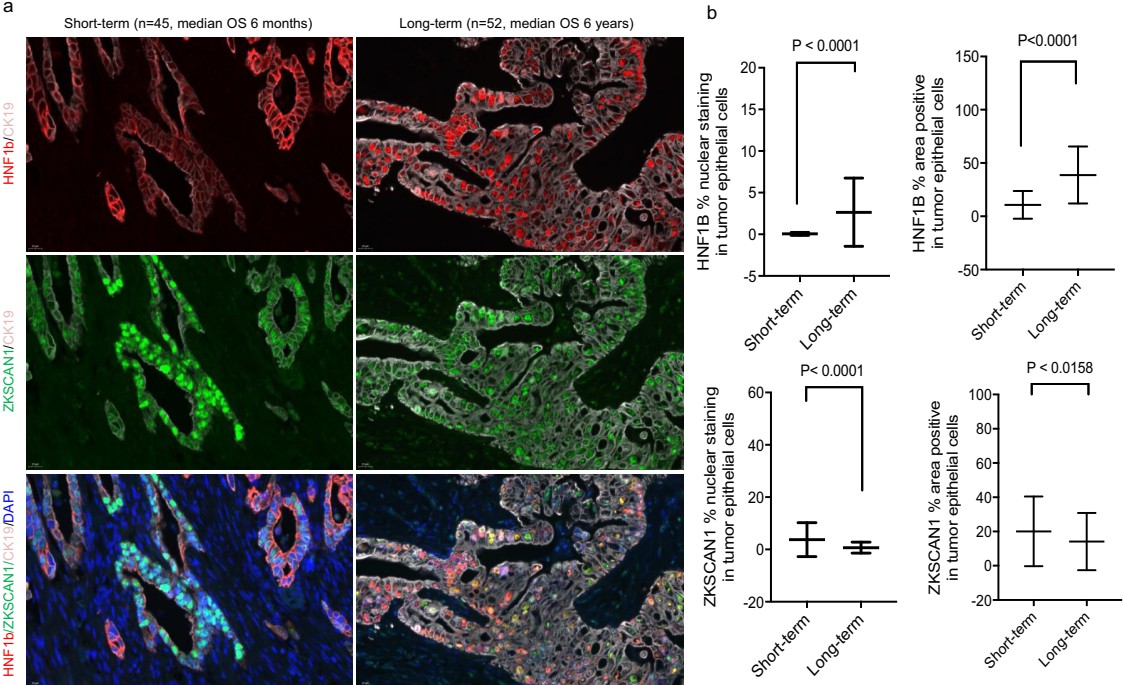

**Fig. 2 Validation of nuclear localization of HNF1b and ZKSCAN1 on tissue microarray from an independent PDAC cohort ($n=97$) selected for long- and short-term survival. a** Staining of HNF1b (red) (upper panel), ZKSCAN1 (green) (middle panel), and combined signal of these two TFs with DAPI (blue) and CK19 (gray) (lower panel), in short-term and long-term survivors. Scale bars are 20 µM as displayed at the left bottom corners of all the micrographs. **b** Quantitation of cells displaying nuclear staining as well as total staining (area positive) for HNF1b (upper panel), and ZKSCAN1(lower panel). Data represented as mean ± SD. Statistical tests are unpaired two tailed t-test with $P < 0.05$ is significant, comparing short-term ($n=45$) and long-term ($n=52$) survivor patients (Source data are provided as a Source Data file).

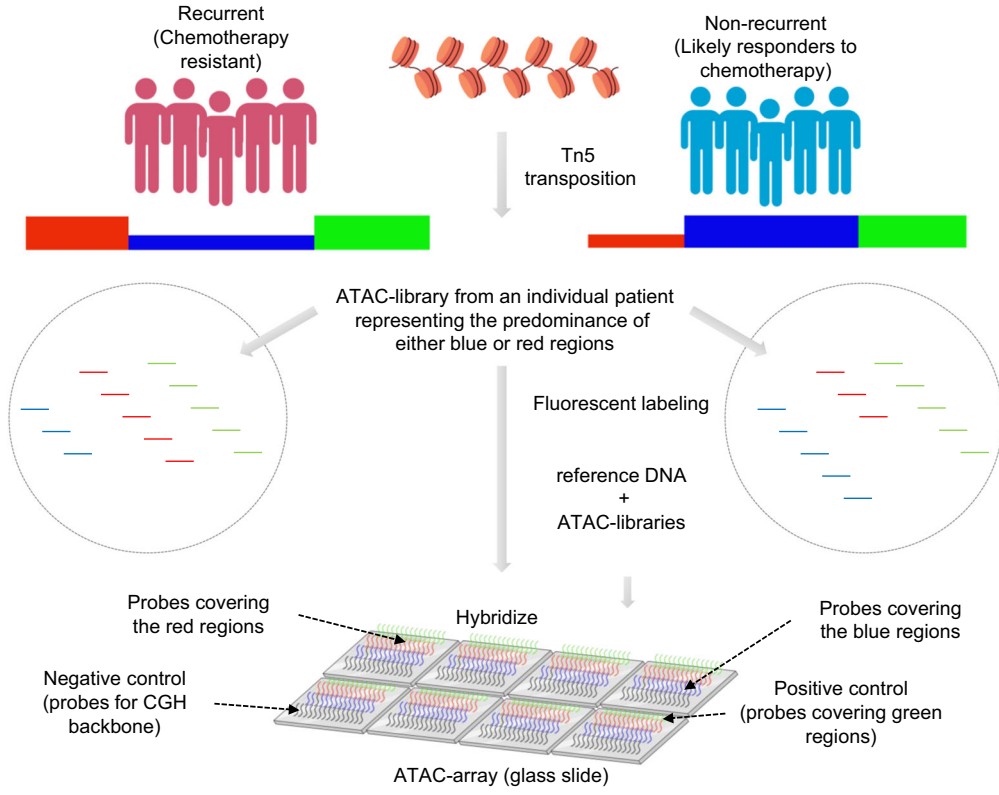

**Fig. 3 ATAC-array design.** Schematic diagram showing the design of the ATAC-array (illustrations were done using BioRender (drawn by S.D.).

distributions by their median values, as exemplified by ATAC-array output results in two representative patients with good prognosis (PT67) and with bad prognosis (PT60), as shown in Fig. 4b (left) and 4b (right) respectively. The Green distribution represents the positive control probes (median value denoted by CTRL) covering 312 regulatory regions open in all PDAC tumors; Black represents the negative control comprising over 7000 probes covering the CGH-backbone as provided by Agilent (median value, CGH); Blue comprises the 688 regulatory regions open in patients with good prognosis (median value, BLUE); and Red comprises of 244 regulatory regions open in patients with poor prognosis (median value, RED). We compared the discriminative value of the BLUE and RED scores individually, as well as that of the difference in distribution median values, (BLUE–RED), all normalized by the difference between positive and negative control distribution medians (CTRL-CGH). For each individual patient, we calculated the ratios of (BLUE/(CTRL-CGH)), (RED/(CTRL-CGH)), and ((BLUE–RED)/(CTRL-CGH)), and found that the score (BLUE/(CTRL-CGH)) displayed the best performance for stratifying patients according to prognosis (Supplementary Fig. 7, RED/(CTRL-CGH) log-rank (Mantel–Cox) test $P = 0.44$, HR 0.77, 95% CI 0.3943 to 1.504, median DFS 559 days ($n = 25$), and median DFS 592 days ($n = 24$) respectively; and (BLUE–RED)/(CTRL-CGH) log-rank (Mantel–Cox) test $P = 0.12$, HR 1.771, 95% CI: 0.8556−3.664, median DFS 663 days ($n = 22$), and median DFS 348 days ($n = 21$) respectively. In particular, we separated patients into two groups using the median value of (BLUE/(CTRL-CGH)) (median = 0.6, range=0.36 to 0.88), which we called the "Prognosis Score" (Supplementary Fig. 8a, and Supplementary Data 6), and compared their DFS by Cox proportional hazards regression. With a median 4.15 (min = 3.18, max= 4.75)-year follow-up among our original discovery cohort ($n = 49$), Kaplan–Meier survival analysis showed a significant segregation of the two groups (Fig. 4c, log-rank (Mantel–Cox) test $P = 0.0022$, Gehan–Breslow–Wilcoxon test, $P = 0.0009$, HR 2.896,

95% CI 1.426 to 5.878, median DFS 264 and 845 days, respectively).

**ATAC-array prognosis score combined with HNF1b nuclear localization.** The 3.2-fold difference in DFS based on ATAC-array prognosis score was further increased to 7.4-fold when the ATAC-array score was combined with immunohistochemical HNF1b nuclear localization as an additional biomarker (Supplementary Data 7 and Fig. 4d, log-rank (Mantel–Cox) test $P <$ 0.0001, Gehan–Breslow–Wilcoxon test $P = 0.0004$ and log-rank test for trend $P < 0.0001$). We found that 38.4% of patients (15/39) displayed an ATAC-array good prognosis signature (Prognosis Score higher than the median) in combination with HNF1b localized to nuclei, with median DFS 1343 days; 12.8% (5/39) displayed an ATAC-array good prognosis signature but no nuclear localization of HNF1b, with median DFS 940 days; 28.2% (11/39) showed an ATAC-array poor prognosis signature (Prognosis Score lower than the median) but positive nuclear localization of HNF1b, with median DFS 559 days; and the remaining 20.5% (8/39) showed an ATAC-array poor prognosis signature and no nuclear localization of HNF1b, with median DFS 183 days. Thus, two simple prognostic methodologies (ATAC-array and immunohistochemical determination of HNF1b nuclear positivity), both derived from our ATAC-seq analysis of chromatin accessibility signatures in resected pancreatic cancer, combine to stratify patients into prognostic groups with more than 7-fold differences in DFS.

**Validation of ATAC-array prognosis score on PDAC organoids.** In order to validate our ATAC-array results on an independent validation cohort, we created ATAC libraries from patient-derived PDAC organoids, representing cultures of enriched malignant epithelial cells derived from individual patients[19,20]. In an initial comparison of ATAC-array chromatin

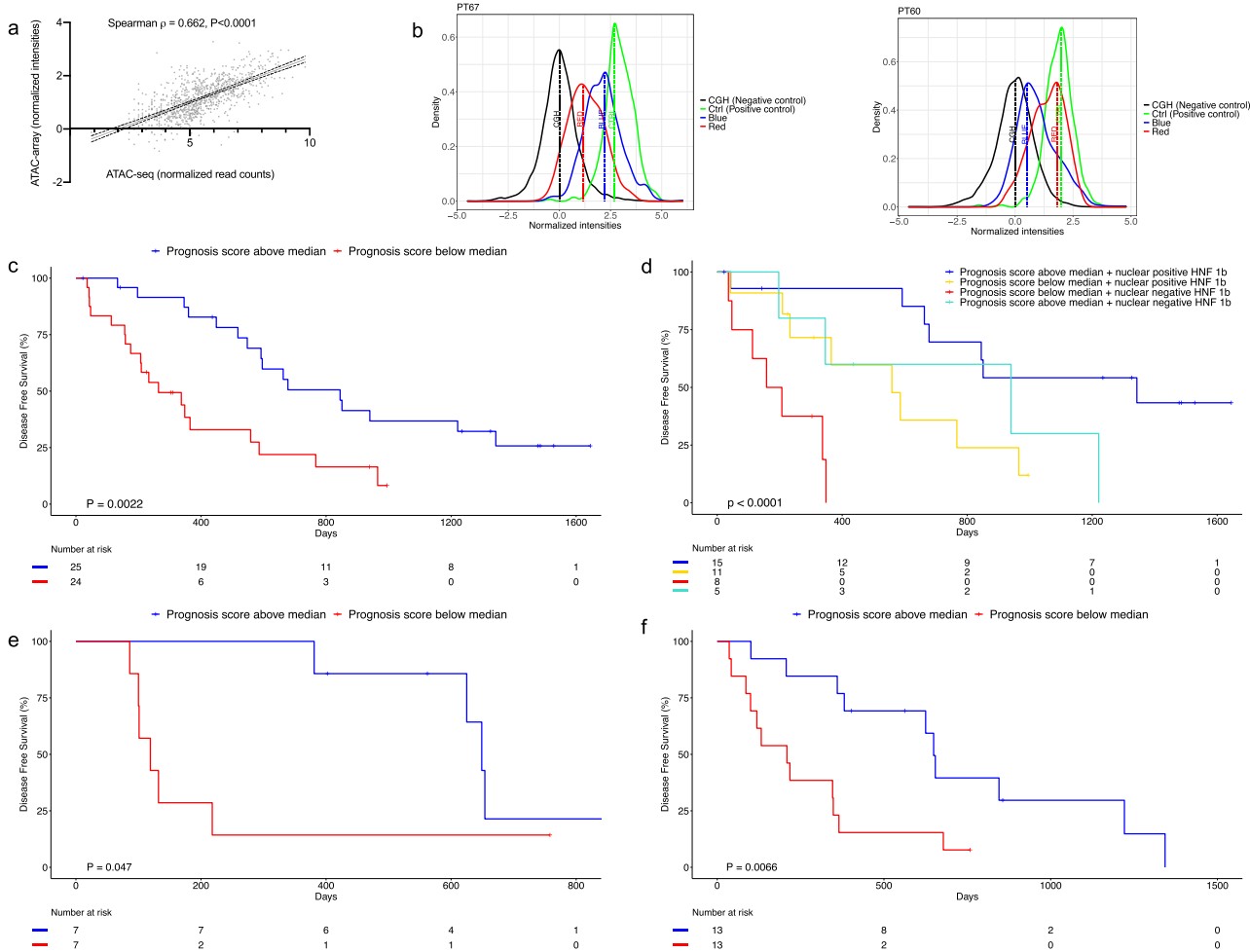

**Fig. 4 ATAC-array correlates with ATAC-seq and predicts DFS. a** ATAC-seq and ATAC-array correlation is shown in a representative patient (PT17). **b** Representative histograms showing good (blue distribution median intensity > red) prognosis and poor (red distribution median intensity > blue) prognosis ATAC-array signature in patient PT67 and PT60, respectively. (Source data are provided as a Source Data file). **c** Kaplan–Meier curve showing significant segregation of PDAC patients ($n = 49$) on the basis of ATAC-array prognosis score (log-rank (Mantel–Cox) test $P = 0.0022$, HR 2.896, 95% CI 1.426 to 5.878). **d** Kaplan–Meier curve shows combination of ATAC-array and HNF1b nuclear localization segregates PDAC patients into four different groups with significantly different median DFS (log-rank (Mantel–Cox) test $P < 0.0001$, and log-rank test for trend $P < 0.0001$). **e** Kaplan–Meier curve showing significant segregation of PDAC organoids on the basis of ATAC-array Prognosis Score in an independent validation cohort ($n = 14$) (log-rank (Mantel–Cox) test $P = 0.0475$, HR 3.228, 95% CI 0.8523 to 12.23). **f** Kaplan–Meier curve showing significant segregation of PDAC organoids on the basis of ATAC-array Prognosis Score in the pooled cohort ($n = 26$) (log-rank (Mantel–Cox) test $P = 0.0066$, HR 2.860, 95% CI 1.144 to 7.145) (Source data are provided as a Source Data file).

accessibility signatures between organoids and freshly isolated EpCAM$^+$ tumor epithelial cells in 12 patients for which libraries were available from both (Supplementary Data 8), we observed significant changes in chromatin accessibility in organoids compared to their tumors of origin, likely representing predictable epigenetic reprogramming of tumor cells occurring during organoid culture. These changes most frequently involved increased accessibility of the "Blue" and "Green" chromatin loci and decreasing accessibility of "Red" regions in organoids (Supplementary Fig. 8b–d). We found that even after taking organoid culture-induced epigenetic alterations into consideration, the Prognosis Score as estimated by ATAC-array on each organoid significantly correlated with the actual DFS of each patient (Spearman $\rho = 0.657$, 95% CI 0.1150 to 0.8978, $P = 0.0238$, $n = 12$, Supplementary Fig. 8e). We additionally analyzed chromatin accessibility in 14 organoids from an independent validation cohort from MSKCC, derived from resected PDAC patients treated with adjuvant Gemcitabine (Supplementary Data 9). As shown in Fig. 4e, when this cohort was separated into two groups using the median Prognosis Score (median = 0.86, range=0.66 to 1.04), Kaplan–Meier survival analysis confirmed a significant segregation in DFS (log-rank (Mantel–Cox) test P = 0.0475, Gehan–Breslow–Wilcoxon test, $P = 0.0080$, HR 3.228, 95% CI 0.8523 to 12.23, median DFS 119 and 649 days respectively). When organoids from both groups were pooled together to create a larger cohort (12 + 14 = 26) as shown in Fig. 4f, the segregation of the organoids on the basis of the Prognosis Score (median = 0.84, range=0.48–1.22, Supplementary Fig. 8a) was statistically more significant (log-rank (Mantel–Cox) test $P = 0.0066$, Gehan–Breslow–Wilcoxon test, $P = 0.0039$, HR 2.860, 95% CI 1.144 to 7.145, median DFS 209 and 649 days, respectively).

## Discussion

In summary, we identified a chromatin accessibility signature and associated TFs that were significantly correlated with PDAC

prognosis, offering a new chromatin organization-based prognostic paradigm for precision oncology. Although chromatin accessibility patterns have been reported in malignant diseases based on epigenetic analyses of bulk tumors[21], to date these analyses have excluded pancreatic cancer, based upon the notoriously low cellularity of these tumors. Our results suggest that tumor-intrinsic chromatin accessibility patterns of PDAC and associated nuclear localization of TFs may predict outcome in this disease. Whether adverse patterns of chromatin accessibility might assist in the selection of PDAC patients for epigenetic "reprogramming" therapy[22] remains to be determined. Nevertheless, we suggest that our ATAC-array technology, combined with immunohistochemical determination of HNF1b nuclear localization, provides a simple and clinically achievable prediction of favorable vs unfavorable epigenetic states in this disease.

## Methods

**Patient recruitment**. All tissues were collected at MSKCC following a study protocol approved by the MSKCC Institutional Review Board. Informed consent was obtained from all patients. Patient samples were collected starting from Sept 2015 to March 2017 and followed until Nov 2017 for the discovery analysis, and until July 2020 for the full cohort (Supplementary Data 2). The study was in strict compliance with all institutional ethical regulations. All tumor samples were treatment-naïve surgically resected primary PDACs. Patients treated with neoadjuvant therapy were excluded. Only histologically confirmed PDAC tumors were included in the study.

**Sorting of tumor cells by EpCAM antibody-conjugated magnetic beads**. We established single-cell suspensions of surgically resected tumors by taking a small piece of PDAC tumor tissue collected in a cell-dissociation media [5 ml of minimal essential media (MEM) containing 100 µl of Liberase-TM (2.5 mg/ml stock solution Roche/Sigma Aldrich Cat# 5401046001), 50 µl of Kolliphor® P 188 (15 mM stock solution Sigma Cat# K4894), 37.5 µl of 1 M CaCl₂ and 5 µl of DNAse-1 (Sigma Cat# DN25 1000X stock 10 mg/ml)]. Single-cell suspensions were then established using a Gentle-MACS tissue dissociator (Miltenyi Biotech) following the manufacturer's guideline (1 h of gentle dissociation of tissues at 37 °C). The viable (>90%viability) single cells were then incubated with EpCAM-conjugated magnetic beads (Miltenyi Biotech Cat# 130-061-101) and then sorted in a magnetic field. EpCAM⁻ cells (in the effluent) were also collected as controls.

**Genome-wide open chromatin profile by ATAC-seq**. Two aliquots of 50,000 EpCAM⁺ cells were taken for ATAC-seq library preparation following a method as described by Buenrostro et al.[10]. Suspensions of 50,000 cells were first pelleted by centrifugation and then washed once with PBS followed by gentle (no rigorous vortex) resuspension in ATAC-seq lysis buffer {10 mM Tris·Cl, pH 7.4, 10 mM NaCl, 3 mM MgCl₂ and 0.1% (v/v) Igepal CA-630, 0.1% Tween-20 (100 µl 100X Tween 20 from a 10% stock solution) and 0.01% Digitonin (100X Digitonin from a stock solution of 1%) was added before use} in order to retrieve healthy nuclei from the cells. TN5 transposase was added to the buffer solution (Nextera DNA-library preparation kit, Illumina, Cat# FC-121-1030, modified with the addition of 0.1% Tween-20 (100 µl 100X Tween 20 from a 10% stock solution) and 0.01% Digitonin (100X Digitonin from a stock solution of 1%) before use) and incubated at 37 °C for 30 min. After the incubation, the transposed DNA fragments were extracted from the reaction solution using the Mini Elute PCR purification kit (Qiagen Cat# 28004) and then amplified by a 12-cycle PCR amplification step with specific primers ordered from IDT (Supplementary Data 10) as described by Buenrostro et al.[10]. The duplicate libraries were then sequenced by paired end, 50 base pair sequencing on an Illumina HiSeq 2500 with an average read depth of 80 million reads per library.

**DNA/RNA extraction**. DNA and RNA were extracted from the same samples of each of the EpCAM⁺ and EpCAM⁻ subpopulations using Qiagen All-prep DNA/RNA micro kit (Qiagen Cat # 80284) following the manufacturer's standard guidelines.

**Quantitation of mutant allele frequencies of the panel of 20 driver genes in PDAC**. Using approximately 50 ng of the extracted genomic DNA, we performed TruSeq Custom Amplicon v2.0 (Illumina) targeted re-sequencing experiments on a selected set of pancreatic cancer driver genes using a custom pancreatic cancer panel. The custom pancreatic cancer panel was established using Illumina TruSeq Amplicon-Cancer Panel platform which provided custom-designed, optimized oligonucleotide probes for sequencing mutational hotspots of pancreatic cancer in >117 kilobases (kb) of target genomic sequence. Within this highly multiplexed, single-tube reaction, 20 genes are targeted with 1242 Amplicons. Each amplicon

had one pair of oligos designed to hybridize to the region of interest. The reaction was then followed by extension and ligation to form DNA templates consisting of regions of interest flanked by universal primer sequences. These DNA templates were amplified by indexed primers and then pooled into a single tube in order to sequence on an Illumina MiSeq sequencing machine. Canonical variant alleles for KRAS and TP53 were preselected from TCGA and ICGC mutation databases as the most frequently recurrent hotspot variant alleles in PDAC.

**Transcriptome analysis**. We performed transcriptome analysis of EpCAM⁺ and EpCAM⁻ cells using the 3′-end Sequencing (Quant-seq) method as described elsewhere[11]. For bulk tumors analyzed by RNA-seq analysis, FASTQ files were aligned using STAR (v2.5.0b, default parameters)[23] to the hg19 genome assembly. Read counting was performed using htseq-count (v0.9.1, parameters: --stranded=no -t exon)[24]. Differential expression was conducted using DESeq2 (v1.18.0)[25].

**ATAC-seq analysis**. Raw FASTQ files were first trimmed using trimmomatic (v0.35, Parameters: TruSeq3-PE adapters, LEADING:3 TRAILING:3 SLIDINGWINDOW:4:15 MINLEN:36)[26]. The samples were then aligned to hg19 genome using bowtie2 (v2.2.6, Parameters: -X2000 –local –mm --no-mixed --no-discordant)[27]. Duplicate read removal was performed using MarkDuplicates (v2.9.0) (Picard Tools—Accessed October 2, 2018. http://broadinstitute.github.io/picard/). In order to account for Tn5 shift, all positive strand reads were shifted by +4 bps and all negative strand reads were shifted by -5bps. Peak calling was then performed on each of the libraries individually and after pooling replicates using MACS2 (v2.1.0, parameters: --nomodel --extsize 150 --shift -75 --slocal 5000 --llocal 20000 -B --SPMR --keep-dup all -p 0.01)[28]. Finally, IDR (irreproducible discovery rate)[29] was used to identify reproducible peaks from the duplicate libraries for each sample (IDR < $1 \times 10^{-2}$). 14 patients from the bottom quartile of reproducible peaks were excluded to select the best quality samples. After identification of reproducible peaks, an atlas of peaks was created from all samples using custom scripts, as previously described elsewhere[30]. Briefly, a simple heuristic was used to combine peaks from multiple cell types to build a common atlas comprising peaks from all patients. The set of non-overlapping peaks from the first two patients was added to the atlas. Then the following heuristic was used to combine overlapping peaks: if the overlap between the peaks was >75%, the non-overlapping portions of the peaks are removed to create a single unified peak which is added to the atlas; if the overlap is <75%, the overlapping portions are removed to create two separate peaks which are both added to the atlas. This procedure was extended to all the patients to create the atlas of all ATAC-seq peaks for subsequent analyses. Annotation of peaks was conducted as described previously[30]. Read counting for all peaks in the atlas was performed using GenomicRanges's summarizeOverlaps function[31]. Differential peak analysis was conducted using DESeq2's generalized linear model function.

**Saturation analysis**. To discover cohort-level saturation, all 54 patients were first randomly sampled without replacement 500 times. Then for each instance of a sample, we counted the total number of peaks in the atlas created by iteratively including patients until we reached all 54 patients.

**Motif analysis**. All peaks in the atlas were first scanned with FIMO[32] to find motif matches. CIS-BP database was filtered as described elsewhere[33] and used for motifs. The result was converted into a matrix where each row is a peak in atlas and each column is a binary presence/absence of a TF. This matrix (X), along with the log2 fold change from differential peak analysis between recurrent vs. non-recurrent patients (y), was used in the following ridge regression framework to predict which TF motifs are differentially accessible:

$$\hat{\beta} = \mathrm{argmin}_\beta \|y - X\beta\|^2 + \lambda \|\beta\|^2$$

Glmnet[34] was used to train and optimize the model using 5-fold cross validation. The resulting coefficient vector was plotted.

**Tissue microarray**. Surgical pathology databases of Memorial Sloan Kettering Cancer Center were searched for patients with a diagnosis of PDAC. Resections of 44 out of 54 cases were identified for which the slides and tissue blocks were available. All hematoxylin and eosin (H&E) slides were re-reviewed, and the best representative tumor area was marked for each case. The formaldehyde fixed paraffin embedded tissues (FFPE) corresponding to the selected histological sections were sampled from these marked regions and a tissue microarray (TMA) was created using three 1 mm diameter punches per tumor. Normal pancreatic areas were also labeled for 6 cases (three cores from each) and used as control tissue. Following the TMA preparation and final evaluation, the data obtained from folded /missing cores were excluded. Our final TMA cohort included 43 PDAC patients.

**Immunostaining**. The immune staining was performed at the Molecular Cytology Core Facility of Memorial Sloan Kettering Cancer Center using a Discovery XT processor (Ventana Medical Systems). The tissue sections were de-paraffinized

with EZPrep buffer (Ventana Medical Systems), antigen retrieval was performed with CC1 buffer (Ventana Medical Systems). Sections were blocked for 30 min with Background Buster solution (Innovex), followed by avidin-biotin blocking for 8 min (Ventana Medical Systems). Multiplexed immunostaining was done as previously described[35]. First, sections were incubated with anti-ZKSCAN1 (Sigma, cat#HPA006672, 0.5 μg/ml) for 5 h, followed by 60-min incubation with biotinylated goat anti-rabbit IgG (Vector labs, cat#PK6101) at 1:200 dilution. The detection was performed with Streptavidin-HRP D (part of DABMap kit, Ventana Medical Systems), followed by incubation with Tyramide Alexa Fluor 488 (Invitrogen, cat# B40953) prepared according to manufacturer instruction with predetermined dilutions. Next, sections were incubated with anti-HNF1b (Sigma, cat#HPA002085, 1 μg/ml) for 5 h, followed by 60-min incubation with biotinylated goat anti-rabbit IgG (Vector labs, cat#PK6101) at 1:200 dilution. The detection was performed with Streptavidin-HRP D (part of DABMap kit, Ventana Medical Systems), followed by incubation with Tyramide Alexa 568 (Invitrogen, cat# T20948) prepared according to manufacturer instruction with predetermined dilutions. Finally, sections were incubated with anti-CK19 (Abcam, cat#ab52625, 0.02 μg/ml) for 5 h, followed by 60-min incubation with biotinylated goat anti-rabbit IgG (Vector labs, cat#PK6101) at 1:200 dilution. The detection was performed with Streptavidin-HRP D (part of DABMap kit, Ventana Medical Systems), followed by incubation with Tyramide Alexa 647 (Invitrogen, cat# B40958) prepared according to manufacturer instruction with predetermined dilutions. Slides were counterstained with DAPI (Sigma Aldrich, cat# D9542, 5 μg/ml) for 10 min and mounted with Mowiol and glass coverslip.

**Subjective scoring of immunohistochemistry (IHC) and quantitative analysis of immunofluorescence (IF) staining.** Subjective scoring was done under the light microscope on a 0-3 scale, with 0 = absent, 1 = weak, 2 = moderate and 3=strong staining. For quantitative analyses of IF, slides were scanned with Panoramic Flash (3DHistech, Hungary) using ×20/0.8NA objective, and regions of interest were drawn using Case Viewer (3DHistech, Hungary). The images were then analyzed using Image J/FIJI (NIH) to count cells with ZKSCAN1, HNF1b, and CK19. The DAPI channel was used to obtain the total nuclear content. Applying background subtraction and median filter preprocessing, the images and the masks were obtained by intensity thresholding and water shedding (grayscale). The thresholds of all channels were individually set to adjust the co-localization and then the absolute counts of cells for the combination of channels were measured.

**ATAC-array.** Using Agilent Technologies' custom CGH-array format we arrayed 932 out of 1092 regions from the chromatin accessibility signature (Agilent probes were not available to cover the remaining 160 regions). 244 regions that were open in the recurrent but silenced in the non-recurrent group were designated as "Red" regions; 688 regions that were open in the non-recurrent group but silenced in the recurrent group were designated as "Blue" regions; 312 control regions that were open in both recurrent and non-recurrent groups were designated as "Green" regions; and Agilent's ~7000 aCGH backbone probes served as negative control regions designated as "Black". Then we hybridized the microarray with fluorescent-labeled ATAC libraries along with a reference genome of known copy numbers (control for normalizing the hybridization efficiency in each array). ATAC-array hybridization was done using the Agilent aCGH hybridization protocol. Briefly, the reference genomic DNA with known copy number (Agilent Technologies, catalog # 5190-4370, lot# 0006392634) was labeled with Cy3 and the ATAC libraries were labeled Cy5 using Genomic DNA ULS labeling kit (Agilent Technologies, catalog # 5190-0420). After estimating the labeling efficiencies independently by nanodrop, we mixed the reference gDNA and ATAC libraries together and hybridized with the microarray for overnight following the manufacturer's aCGH hybridization protocol. After 16-24 h of hybridization at 65 °C, we washed the microarray with wash buffers (Agilent Technologies) and scanned the array on a SureScanDx microarray reader (Agilent Technologies). We use reference gDNA (Cy3) as the control to normalize the hybridization efficiencies on each probe. The microarray data were analyzed by standard bioinformatic pipeline.

**ATAC-array analysis pipeline.** The microarray slides were scanned using Agilent's Sure Scan G3 microarray scanner and the resulting probe grid image in TIFF format was scanned by Agilent's commercially available proprietary Feature Extraction (FE) software (FE Version 12.1.0.3) to give the output text files containing the probe intensities. These files were subjected to an analysis pipeline where the probes in every sample are mapped to the desired ATAC differential regions (Blue and Red), control regions and CGH-backbone regions and the probes are classified accordingly. The median normalized intensities of all the probes for all the ATAC differential regions, control regions and CGH backbone regions were outputted along with the gene annotations and the samples are classified as either bad prognosis (red) or good prognosis (blue) based on pre-defined Red/Blue differential regions in the ATAC differential regions file and the median log ratios of these Red/Blue regions along with the differential significance ($P < 0.05$ by t-test)

**Reporting summary.** Further information on research design is available in the Nature Research Reporting Summary linked to this article.

## Data availability
Processed ATAC-seq and RNA Quant-seq data have been deposited to GEO (GSE124229 and GSE124230, respectively). Raw data have been made accessible through controlled-access dbGaP portal (phs002394.v1.p1). Potential users will need to complete and be approved of a data access request and then the raw data can be used according to the terms of the consent, and the data use limitations for the subjects. The ATAC-array raw data are uploaded in the following GitHub link along with the analysis code https://github.com/hchintalapudi/ATAC-seq-ATAC-array. Any other relevant data are available from the authors upon reasonable request. Source data are provided with this paper.

## Code availability
All code is available through the following GitHub link https://github.com/hchintalapudi/ATAC-seq-ATAC-array

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

## Acknowledgements

The authors thank all the patients and their families for their kind co-operation. We thank Mr. Brian Herbst and Ms. Jacklyn Egger at the MSKCC for helping us with the collection of tumor tissues and cryosectioning of the frozen tumor specimens. This work was funded by the Ring Family Foundation, by NCI R01CA204228 (to SDL), NCI U01CA224175 (to VPB and SDL), and AACR/Pancreatic Cancer Action Network Research Acceleration Grant 344673 (to SDL). We also wish to acknowledge support from the Dartmouth NCCC Cancer Center Support Grant (NCI P30 CA023108, the MSKCC Cancer Center Support Grant (NCI P30CA008748), services provided by Norris Cotton Cancer Center Genomics and Biostatistics Shared Resources, and the Integrated Genomics and Immunohistochemistry Core at MSKCC.

## Author contributions

S.D. designed the study, performed and coordinated the experiments, and drafted the manuscript. S.C. designed and performed the bioinformatic analyses and contributed to writing the manuscript. B.C. and H.C. helped in ATAC-array bioinformatic analyses. V.A., A.L.M., and D.T. contributed to wet-lab experiments. A.G. provided the histopathological evaluations. L.Z. and A.M.-M. performed the Amplicon targeted re-sequencing experiments. N.L., J.P.M., and J.B. contributed to developing the organoids. R.N. and R.M. performed RNA analysis. S.L. supervised the ATAC-array design and manufacture by Agilent. S.L. helped to set up the ATAC-array experiment in the laboratory. S.S., A.C., D.G., and K.Y. provided the evaluation of clinical records. J.G. provided the statistical oversight. C.I.D. provided the organoids, and V.P.B. provided the short- and long- term PDAC cohort. R.C. optimized the ATAC-seq technique. C.S.L. supervised the bioinformatic analyses and contributed to writing the manuscript. S.D.L. conceived and supervised the study and contributed to writing the manuscript.

## Competing interests

ATAC-array technology has been filed under U.S. patent PCT/US2019/046301. Patent applicants are the Trustees of Dartmouth College, 11 Rope Ferry Road, New Hampshire, 03755, US, and Memorial Sloan Kettering Cancer Center, 1275 York Avenue, New York, New York, 10065, US. Inventors, Dhara, Surajit, Leach, Steven D., Chhangawala, Sagar, Leslie, Christina. Parts of the manuscript explaining the development of the ATAC-array technology and also the prediction of DFS (following gemcitabine adjuvant chemotherapy) by combining ATAC-array and HNF1b of this patent have been included in this patent. The international patent is published on February 20, 2020 (WO/2020/036929). Surajit Dhara and Steven D. Leach are co-founders of Episteme Prognostics; work on this project has been performed under a formal Conflict of Interest Management Plan at Dartmouth College. Steven D. Leach is also a member of the Scientific Advisory Board of Nybo Therapeutics. Other authors declare no competing interests.
