## [Peer Review File · Nature Communications]

Reviewers' Comments:

Reviewer #1:

Remarks to the Author:

Dhara et al. used ATAC-seq on 54 pancreatic ductal adenocarcinoma (PDAC) samples to determine chromatin accessibility differences between patients with and without relapse at one year. They found 1,092 differentially accessible peaks, which were enriched for the motifs of two transcription factors (ZKSCAN1 and HNF1b). They also developed a chromatin accessibility microarray technology (ATAC-array) to determine chromatin accessibility, as a cheaper alternative to ATAC-seq, and validated it on an independent cohort. By using the results from ATAC-array, they were able to develop a PDAC prognosis score, which discriminates between short and long disease-free survival of PDAC patients.

I have one major concern with this study. The role of HNF1b in pancreas and pancreatic cancer is well known and it has been shown to be a diagnostic marker for PDAC (PMC6062876), but the authors describe the findings about this transcription factor as novel. These are just confirmation of the function of HNF1b as a tumor suppressor in PDAC (PMC4247517; PMC6722301 and others). Reading the second part of the manuscript, where the authors describe the ATAC-array-based prognostic score, it is unclear what information it provides in addition to the HNF1b staining results? I mean: if the authors divide the samples into those that have high and low expression of HNF1b, would they observe the same trends as the high and low prognosis score? If the authors fail to show that the prognosis score is more accurate than HNF1b staining, much of the novelty of this manuscript would be lost.

Other major comments:

1. Supplementary tables that contain data should be provided in excel format, not in PDF, which cannot be used by the reader.
2. Is there any difference in the proportion of malignant epithelial cells between samples with short and long disease-free prolonged survival (DFS)? This should be tested and clearly stated in the text, as it may be a confounding factor for the conclusions that the authors make. If there is a difference, it should be accounted for in the differential chromatin accessibility analysis as a covariate.
3. The ATAC-seq differential expression analysis lacks detail ("ATAC-seq analysis" in the Methods section, page 15). Specifically:
 - o "After identification of reproducible peaks, an atlas of peaks was created from all samples using custom scripts". The authors provide some detail in the Results section, but it is unclear how peaks from different patients were merged. For example: do the authors require a certain overlap between peaks called in different patients to include a peak in their "atlas"?
 - o Importantly, sex and age (and in general, patient information) of the patients is not provided. Please, provide this information as a supplemental table. This information is essential for the differential peak analysis, as certain characteristics (at a minimum sex and age, but also the fraction of mitochondrial reads, which is not provided either and is usually an issue in ATAC-seq experiments) should be included as covariates in the GLM that DESeq2 uses. Were these covariates included in the analysis?
4. It would be interesting to directly test the association between DFS and chromatin accessibility, in addition to the binary test that the authors performed. This analysis would result in the identification of peaks that become increasingly stronger (or weaker) as a function of the disease-free survival time. It could potentially identify a more sensitive list of peaks.
5. I do not agree with the conclusion of the first paragraph of page 7: "These studies demonstrate the utility of using unbiased assessment of chromatin accessibility to identify new prognostic biomarkers in human pancreatic cancer". The results described in this paragraph show that HNF1b

protein expression and localization is different between samples with short and long DFS.

6. Prognosis score should separate patients with short and long DFS. However, the authors just describe that the score (BLUE/(CTRL – CGH)) provides the best performance but there is no data supporting this statement (bottom of page 8). Looking at the examples in Figure 4B, the simple ratio between blue and red would be good and easy to explain, as the score used by the authors completely removes peaks open in samples from patients with bad prognosis. There are many ways to test this and to test that the prognosis score is meaningful. Given the small sample set, probably a leave-one-out cross-validation would be a good approach.

Potential analyses to strengthen the last part of the paper, which seems pretty weak, with very little validation sets (and not even primary tumors).

1. The authors could use publicly available data, such as from the TCGA (<https://science.sciencemag.org/content/362/6413/eaav1898>: this paper should be cited in the introduction too). Although it does not have ATAC-seq from pancreatic cancer, it can be used to expand the results from this manuscript and maybe it could show that the prognosis score can be generalized to other tumor types.

Minor comments:

1. Figure 1B: it would be too confusing to specify the Y axis on all the tracks, but the legend should provide a description of it. At least that all the track lines have the same Y axis limits and that the peak height is normalized by the total read depth of each sample.

2. The validation sets are small and they are from organoids, rather than primary tumors. Can the authors find publicly available ATAC-seq data on pancreatic cancers?

Reviewer #2:

Remarks to the Author:

In the manuscript submitted by Dr. Leach, the authors used ATAC-seq based method, to find makers that will help to predict PDAC patient's survival after tumor resection. The approach is interesting and seems to be successful in predicting patient survival when being used on an independent validation cohort. The study can be extended to further investigate additional predictions from the same data and be used to find new therapies. In summary, this is very important work adding a new diagnostic tool, where it is very much needed. I believe the authors can extend the analysis to improve their tool.

I have several concerns regarding the manuscript:

1) The author used a cohort of 54 samples for the initial statistical analysis and the measurements of signal to noise ratio. The analysis finding of DE TF is based on 16 samples (6 recurrent and 10 non-recurrent). Can the authors add the panels in Fig S1, S2, and S3 showing only the 6 and 10 samples that were considered for the analysis in Fig. 1C.

2) In light of the previous point, it will be interesting to estimate the rate of false-negative results, therefore, highlighting a potential benefit from a similar study with a larger number of patients. In addition, a better EpCAM+ cell purification strategy is expected to produce stronger signals.

3) It is critical to examine if adding more significant TF such as DNMT1 and EPAS1 will improve the predictions of patient survival. The authors use the most significant hit but adding TF can improve the results.

4) Information on the biological role of ZKSCAN1 and HNF1b is missing. Also, which genes are being controlled by these TFs. This information is needed to achieve a better understanding of the biology of PDAC, but may also have a clinical impact, for example, if some of the genes that are being regulated by these TFs, encode secreted proteins that can be detected in the blood before surgery.

5) It is clear that the authors aim to produce a method that is cheap and can be done in a clinic that is not fully equipped. However, it is not clear to what extent ATAC-array is better than ATAC-seq, sample purification is similar and sequencing services are available everywhere these days.

RESPONSE TO REVIEWER COMMENTS (reviewer comments in grey; our responses in black)

Dear Reviewers,

The authors wish to thank the Editor and both Reviewers for their encouragement and many insightful comments. Indeed, we were very pleased to learn that the reviewers felt that our work was “very important” and “very much needed”. Even more importantly, we are very grateful for the wonderfully helpful comments and suggestions provided by the reviewers, and are now honored to submit a much improved manuscript which we hope will be deemed ready for publication in *Nature Communications*. As requested, the revised manuscript includes both additional methodologic detail as well as extensive new data. These changes are summarized in the following Response to Reviewer Comments, in which we address each and every reviewer concern on a point-by-point basis.

Again, we are very thankful to the reviewers for helping us to improve our work, and grateful for the chance to publish in *Nature Communications*.

Reviewer #1 (Remarks to the Author):

Dhara et al. used ATAC-seq on 54 pancreatic ductal adenocarcinoma (PDAC) samples to determine chromatin accessibility differences between patients with and without relapse at one year. They found 1,092 differentially accessible peaks, which were enriched for the motifs of two transcription factors (ZKSCAN1 and HNF1b). They also developed a chromatin accessibility microarray technology (ATAC-array) to determine chromatin accessibility, as a cheaper alternative to ATAC-seq, and validated it on an independent cohort. By using the results from ATAC-array, they were able to develop a PDAC prognosis score, which discriminates between short and long disease-free survival of PDAC patients.

I have one major concern with this study. The role of HNF1b in pancreas and pancreatic cancer is well known and it has been shown to be a diagnostic marker for PDAC (PMC6062876), but the authors describe the findings about this transcription factor as novel. These are just confirmation of the function of HNF1b as a tumor suppressor in PDAC (PMC4247517; PMC6722301 and others). Reading the second part of the manuscript, where the authors describe the ATAC-array-based prognostic score, it is unclear what information it provides in addition to the HNF1b staining results? I mean: if the authors divide the samples into those that have high and low expression of HNF1b, would they observe the same trends as the high and low prognosis score? If the authors fail to show that the prognosis score is more accurate than HNF1b staining, much of the novelty of this manuscript would be lost.

Author Rejoinder: First, we thank the reviewer for their kind review and feedback on the manuscript. In particular, we are grateful to the reviewer for pointing out the reference regarding HNF1b as a prognostic marker for PDAC and other malignancies. This background information is now incorporated in the manuscript (page 6).

In our 1092 signature we identified 723 regulatory genomic regions, including the putative promoter region of HNF1b, which were silenced in poor prognosis PDAC patients. Our ATAC-array prognosis score is derived from summing up all these silenced regions (BLUE).

Of course, HNF1b nuclear localization and the ATAC-array prognosis score both independently segregated patients into two different DFS groups as depicted in **Fig 1e** and **Fig 4c** respectively. However, as shown in **Figure 4d**, we can even more effectively subclassify patients into 4 sub-groups when we combine HNF1b and the ATAC-array prognosis score together. This is exemplified by the fact that there was a group of patients (17.9%) with a high ATAC-array prognosis score (> 0.6 median value, which predicts good prognosis) who did *not* have HNF1b localized in their nuclei (which is the sign of bad prognosis), and also vice versa (15.4%). These two “intermediate” groups displayed corresponding intermediate DFS of 940 and 767 days respectively. In contrast, there were two “extreme” groups: 33.3% of patients had both a high ATAC-array prognosis score (> 0.6) and HNF1b localized to nuclei (both indicative of good prognosis) and showed a median DFS of 1343 days, while 33.3% of patients had a low ATAC-array poor prognosis signature (< 0.6) and no nuclear localization of HNF1b, with a median DFS of 337 days.

Therefore, although HNF1b is an independent prognostic biomarker (Fig 1e, $p < 0.01$) the combination score is not only a better marker (Fig 4c, $p < 0.002$) but also further refines the segregation of patients into “good”, “intermediate” and “poor” prognosis groups, with up to 4-fold differences in DFS.

Other major comments:

1. Supplementary tables that contain data should be provided in excel format, not in PDF, which cannot be used by the reader.

Author Rejoinder: We are now resubmitting the tables in excel format. Initially, the journal requirement for these uploads were only in PDF format, but we agree that the excel format will be more useful to readers.

2. Is there any difference in the proportion of malignant epithelial cells between samples with short and long disease-free prolonged survival (DFS)? This should be tested and clearly stated in the text, as it may be a confounding factor for the conclusions that the authors make. If there is a difference, it should be accounted for in the differential chromatin accessibility analysis as a covariate.

Author Rejoinder: We applied Cox regression to evaluate the confounding effect of age, sex and cellularity on DFS. Since none of them showed a significant p-value (less than 0.05), we concluded that there was no significant confounding effect. We also plotted the Kaplan-Meier curves after a median 4.15 years of follow up on the discovery set patients ($n=16$) adjusting for age, sex and cellularity (KRAS variant allele frequency) as displayed in **Supplementary Figure S5a**. The adjusted Kaplan-Meier curves looked almost identical to the unadjusted curve (data

not shown). We have also shown that there is no significant difference in KRAS and TP53 variant allele frequencies or EpCAM and KRT19 mRNA expression between the recurrent (n=6) and non-recurrent (n=10) groups (**Supplementary Figure S5b and c**).

3. The ATAC-seq differential expression analysis lacks detail (“ATAC-seq analysis” in the Methods section, page 15). Specifically:

- o “After identification of reproducible peaks, an atlas of peaks was created from all samples using custom scripts”. The authors provide some detail in the Results section, but it is unclear how peaks from different patients were merged. For example: do the authors require a certain overlap between peaks called in different patients to include a peak in their “atlas”?
- o Importantly, sex and age (and in general, patient information) of the patients is not provided. Please, provide this information as a supplemental table. This information is essential for the differential peak analysis, as certain characteristics (at a minimum sex and age, but also the fraction of mitochondrial reads, which is not provided either and is usually an issue in ATAC-seq experiments) should be included as covariates in the GLM that DESeq2 uses. Were these covariates included in the analysis?

Author Rejoinder: Our methodology for analysis is now provided in detail, along with citing the reference (Page 15). As requested by the reviewer, we also have now incorporated age and sex information in the supplementary table S2 (for the patient cohort n=54). With respect to mitochondrial reads, we found that these displayed an inverse correlation with the nuclear reads in each ATAC-seq library (Spearman $\rho=-0.273$, 95% CI -0.439 to -0.088, $P=0.0044$, n=108). Therefore, our regression analysis, which included nuclear reads as a covariate in the GLM, automatically adjusts for the confounding effect of the mitochondrial reads from the discovery of our 1092 signature. This is explicitly mentioned in the manuscript (page 5).

4. It would be interesting to directly test the association between DFS and chromatin accessibility, in addition to the binary test that the authors performed. This analysis would result in the identification of peaks that become increasingly stronger (or weaker) as a function of the disease-free survival time. It could potentially identify a more sensitive list of peaks.

Author Rejoinder: We completely agree with the point. It would have been indeed interesting to perform a non-binary test to discover the direct quantitative association between the peaks and the DFS but unfortunately, we do not have a sample size large enough to perform this regression as the reviewer has suggested.

5. I do not agree with the conclusion of the first paragraph of page 7: “These studies demonstrate the utility of using unbiased assessment of chromatin accessibility to identify new prognostic biomarkers in human pancreatic cancer”. The results described in this paragraph show that HNF1b protein expression and localization is different between samples with short and long DFS.

Author Rejoinder: We appreciate the reviewer’s suggestion that we use different language in expressing our conclusion. The conclusion now reads: “These studies demonstrate that the expression and localization of HNF1b protein, a transcription factor identified through unbiased assessment of chromatin accessibility, is different between samples with short and long DFS.”

6. Prognosis score should separate patients with short and long DFS. However, the authors just describe that the score (BLUE/(CTRL – CGH)) provides the best performance but there is no data supporting this statement (bottom of page 8). Looking at the examples in Figure 4B, the simple ratio between blue and red would be good and easy to explain, as the score used by the authors completely removes peaks open in samples from patients with bad prognosis. There are many ways to test this and to test that the prognosis score is meaningful. Given the small sample set, probably a leave-one-out cross-validation would be a good approach.

Author Rejoinder: As we stated in the manuscript, “For each individual patient, we calculated the ratios of (BLUE/(CTRL - CGH)), (RED/(CTRL - CGH)), and ((BLUE - RED)/(CTRL - CGH)), and found that the score (BLUE/(CTRL - CGH)) gave the best performance for stratifying patients by prognosis”. We apologize for not providing the data supporting this statement. These data are now provided in **Supplementary Figure S6**, in which the less discriminative survival curves and less significant p-values for (RED/(CTRL - CGH)), and ((BLUE - RED)/(CTRL - CGH)) are provided.

Potential analyses to strengthen the last part of the paper, which seems pretty weak, with very little validation sets (and not even primary tumors).

1. The authors could use publicly available data, such as from the TCGA (<https://science.sciencemag.org/content/362/6413/eaav1898>: this paper should be cited in the introduction too). Although it does not have ATAC-seq from pancreatic cancer, it can be used to expand the results from this manuscript and maybe it could show that the prognosis score can be generalized to other tumor types.

Author Rejoinder: We would respectfully suggest that there are no publicly available data sets that would allow us to extend our findings further. Perhaps the single most distinguishing feature of our data set is that we performed a tumor-cell-intrinsic chromatin accessibility analysis, which no other publicly available existing data set has ever previously accomplished. Moreover, the Science 2018 ATAC-seq paper performed ATAC-seq on several types of bulk tumors (without separating the malignant cells from the tumor microenvironment cells) but excluded pancreatic cancer. While we understand the reviewer’s concern that we are not able to validate our data on a publicly available data set, we do believe that our data set will set a new standard establishing tumor-cell intrinsic ATAC-seq as the gold standard approach. This is especially important in PDAC, a notoriously difficult tumor type based on heterogeneity within the tumor microenvironment and variable malignant epithelial cellularity in analyzed specimens.

Minor comments:

1. Figure 1B: it would be too confusing to specify the Y axis on all the tracks, but the legend should provide a description of it. At least that all the track lines have the same Y axis limits and

that the peak height is normalized by the total read depth of each sample.

Author Rejoinder: We thank you for the suggestion and have added the description to the legend as advised.

2. The validation sets are small, and they are from organoids, rather than primary tumors. Can the authors find publicly available ATAC-seq data on pancreatic cancers?

Author Rejoinder: Unfortunately, as mentioned above there are no public data available for PDAC ATAC-seq that include tumor-cell-intrinsic chromatin accessibility patterns similar to what we provide in the current study.

Reviewer #2 (Remarks to the Author):

In the manuscript submitted by Dr. Leach, the authors used ATAC-seq based method, to find markers that will help to predict PDAC patient's survival after tumor resection. The approach is interesting and seems to be successful in predicting patient survival when being used on an independent validation cohort. The study can be extended to further investigate additional predictions from the same data and be used to find new therapies. In summary, this is very important work adding a new diagnostic tool, where it is very much needed. I believe the authors can extend the analysis to improve their tool.

Author rejoinder: We are first very grateful for the reviewer's assessment that our work is "very important work" and "very much needed".

I have several concerns regarding the manuscript:

1) The author used a cohort of 54 samples for the initial statistical analysis and the measurements of signal to noise ratio. The analysis finding of DE TF is based on 16 samples (6 recurrent and 10 non-recurrent). Can the authors add the panels in Fig S1, S2, and S3 showing only the 6 and 10 samples that were considered for the analysis in Fig. 1C.

Author Rejoinder: As the reviewer suggested we have compared the variant allele frequencies for KRAS and TP53 (as shown in S1), among the 16 samples (6 recurrent and 10 non-recurrent) and found no significant differences (t-test, $p > 0,05$), as now displayed in **Supplementary Fig. S5b**. Also, based on the reviewer suggestion we have now shown the relative differences in EpCAM and KRT19 gene expression among these 16 samples separately, as displayed in **Supplementary Fig. S5c**.

2) In light of the previous point, it will be interesting to estimate the rate of false-negative results, therefore, highlighting a potential benefit from a similar study with a larger number of patients. In addition, a better EpCAM+ cell purification strategy is expected to produce stronger signals.

Author Rejoinder: We agree with the point that a better EpCAM+ cell purification strategy would be expected to produce stronger signals. Nevertheless, we showed significant enrichments of KRAS variant allele frequencies in our EpCAM+ subpopulation in each patient when compared with the EpCAM- subpopulation from the same patient. Also, although we understand the potential advantages of a larger sample size, we did perform a cohort level saturation analysis as a means of determining sample size adequacy and found “near-saturation” of our discovery of peaks from the cohort of 54 PDAC patients. Thus, while we had only 16 patients with mature survival data in our original differential survival analysis, our atlas of chromatin peaks was established using a much larger group allowing us to approach saturation in peak discovery.

3) It is critical to examine if adding more significant TF such as DNMT1 and EPAS1 will improve the predictions of patient survival. The authors use the most significant hit but adding TF can improve the results.

Author Rejoinder: Yes, we agree with the reviewer, we could include and test more TFs by IHC, but by combining HNF1b and ATAC-array prognosis score together we have already segregated the patients into three major groups: good-, intermediate-, and poor-prognosis, achieving a 4-fold difference in survival between the good- and poor-prognosis groups and spanning the extremes of DFS for resected PDAC in the literature (Conroy et al NEJM 2018). Therefore, including additional TFs might add more information but we are doubtful if this would add any further meaningful segregation of survival. In addition, the tissue microarrays used to generate sections of IHC and IF unfortunately have now been exhausted, meaning that such an effort is no longer technically feasible.

4) Information on the biological role of ZKSCAN1 and HNF1b is missing. Also, which genes are being controlled by these TFs. This information is needed to achieve a better understanding of the biology of PDAC, but may also have a clinical impact, for example, if some of the genes that are being regulated by these TFs, encode secreted proteins that can be detected in the blood before surgery.

Author Rejoinder: Thank you for this helpful suggestion. Additional information on HNF1b and ZKSCAN1 biology has been added in the manuscript as suggested by the reviewer (Page 6). In response to the reviewer’s suggestion, we have also searched in silico the hits for HNF1b and ZKSCAN1 putative binding sites throughout our global atlas of open chromatin peaks (121,697 peaks) and found 9613- and 5339- hits for HNF1B and ZKSCAN1 respectively. As requested by the reviewer, the lists of nearest genes corresponding to peaks containing these TF binding motifs are now displayed in new **Supplementary Table S4**.

5) It is clear that the authors aim to produce a method that is cheap and can be done in a clinic that is not fully equipped. However, it is not clear to what extent ATAC-array is better than ATAC-seq, sample purification is similar and sequencing services are available everywhere these days.

Author Rejoinder: We appreciate this critical question. We maintain that ATAC-array is not a competing modality with ATAC-seq. Rather these are two different technologies with two different purposes. ATAC-seq is an excellent discovery tool to discover new chromatin accessibility signatures from genome-wide investigation, but once we discover the signature, such as our 1092-chromatin accessibility signature in human PDAC, it is much more convenient to use ATAC-array as a diagnostic tool for routine clinical purposes. It is cheaper, faster, and easy-to-use in any cancer clinic, as compared to the ATAC-sequencing analysis which requires sophisticated bioinformatics support and may or may not be available within a clinically actionable time frame. We are therefore respectfully skeptical that ATAC-Seq will ever be deployable clinically, especially given our demonstration of validity for the cheaper and faster ATAC-array methodology.

Reviewers' Comments:

Reviewer #1:

Remarks to the Author:

The authors have satisfactorily addressed concerns.

Reviewer #2:

Remarks to the Author:

Reviewer #2 (Remarks to the Author):

Regarding my previous points

#1: In the new Supplementary Fig. 5b, five out of six recurrent patients had the G12D Kras mutation, compared to two out of ten in the non-recurrent patients. This result is quite striking, maybe KRAS mutation type is the best predictor or at least should be added to the ATAC-seq and HNF1b.

#4: I can accept that the main goal of the paper is to establish a new diagnostic tool. However, in a journal such as Nature Communication, I think the authors should also extend and add mechanistic insight on PDAC biology.

Why ZKSCAN1 and HNF1B accessibility correlate with the chances of patient recurrency?

The authors generated gene expression data and ATAC-seq from the same samples, how these two data set correlate? More specifically from the genes listed in the new Supplementary table S4, which are differentially expressed between recurrent and non-recurrent patients? Which biological pathways may be involved? This analysis based on the data that the authors already generated may lead to a better understanding of the differences in EPCAM+ cells between the groups of patients.

No other comments

RESPONSE TO REVIEWER COMMENTS (reviewer comments in grey; our responses in black)

Reviewer #1 (Remarks to the Author):

The authors have satisfactorily addressed concerns.

Reviewer #2 (Remarks to the Author):

Regarding my previous points

#1: In the new Supplementary Fig. 5b, five out of six recurrent patients had the G12D Kras mutation, compared to two out of ten in the non-recurrent patients. This result is quite sticking, maybe KRAS mutation type is the best predictor or at least should be added to the ATAC-seq and HNF1b.

Author Rejoinder: We thank you for pointing that out. Although we have not incorporated the information in the manuscript, we have carefully analyzed the confounding effect of KRAS mutant allele frequencies on survival of PDAC. As shown in the figure below, we have first checked the frequencies of all the mutant alleles in our cohort and found two mutant alleles G12D (56%) and G12V (31%) of KRAS are predominant (**Fig a**). However, as shown in the **figure b** and **c** below, and consistent with prior reports, there were no significant differences in survival observed in association with either of these mutant KRAS. Therefore, we have not pursued incorporation of KRAS mutation type into our prognostic model.

#4: I can accept that the main goal of the paper is to establish a new diagnostic tool. However, in a journal such as Nature Communication, I think the authors should also extend and add mechanistic insight on PDAC biology.

Why do ZKSCAN1 and HNF1B accessibility correlate with the chances of patient recurrency?

The authors generated gene expression data and ATAC-seq from the same samples, how do these two data sets correlate? More specifically from the genes listed in the new Supplementary table S4, which are differentially expressed between recurrent and non-recurrent patients? Which biological pathways may be involved? This analysis based on the data that the authors already generated may lead to a better understanding of the differences in EPCAM+ cells between the groups of patients.

Author Rejoinder: We appreciate that the reviewer has inquired about mechanistic details, and we completely agree with the fact that the intricate biology behind our findings could be intriguing. However, we appreciate and agree with the Editor's opinion that the experiments necessary to elucidate this biology are beyond the scope of the present manuscript, and that a follow-up project with functional experiments needs to be performed in order to critically answer these questions.

No other comments